# Inhibitors of the PI3K/Akt/mTOR Pathway in Prostate Cancer Chemoprevention and Intervention

**DOI:** 10.3390/pharmaceutics13081195

**Published:** 2021-08-03

**Authors:** Nazanin Momeni Roudsari, Naser-Aldin Lashgari, Saeideh Momtaz, Shaghayegh Abaft, Fatemeh Jamali, Pardis Safaiepour, Kiyana Narimisa, Gloria Jackson, Anusha Bishayee, Nima Rezaei, Amir Hossein Abdolghaffari, Anupam Bishayee

**Affiliations:** 1Department of Toxicology and Pharmacology, Faculty of Pharmacy, Tehran Medical Sciences, Islamic Azad University, Tehran 1941933111, Iran; nazanin.mroudsari@gmail.com (N.M.R.); narimanlashgari1374@gmail.com (N.-A.L.); shaghayeghabaft19@gmail.com (S.A.); fateme_jamali77@yahoo.com (F.J.); pardis.s76@gmail.com (P.S.); kiyana.narimisa@yahoo.com (K.N.); 2Medicinal Plants Research Center, Institute of Medicinal Plants, Academic Center for Education, Culture and Research, Tehran 1417614411, Iran; saeideh58_momtaz@yahoo.com; 3Toxicology and Disease Group, Pharmaceutical Sciences Research Center, Institute of Pharmaceutical Sciences, Faculty of Pharmacy, Tehran University of Medical Sciences, Tehran 1417614411, Iran; 4Gastrointestinal Pharmacology Interest Group, Universal Scientific Education and Research Network, Tehran 1417614411, Iran; 5Lake Erie Collage of Osteopathic Medicine, Bradenton, FL 34211, USA; gloriajackson229@gmail.com; 6Pine View School, Osprey, FL 34229, USA; anushabishayee@gmail.com; 7Research Center for Immunodeficiencies, Children’s Medical Center, Tehran University of Medical Sciences, Tehran 1417614411, Iran; rezaei_nima@yahoo.com; 8Department of Immunology, School of Medicine, Tehran University of Medical Sciences, Tehran 1417614411, Iran

**Keywords:** prostate cancer, PI3K/Akt/mTOR pathway, small-molecule mTOR inhibitors, natural compounds

## Abstract

The phosphatidylinositol 3-kinase (PI3K)/serine-threonine kinase (Akt)/mammalian target of the rapamycin (mTOR)-signaling pathway has been suggested to have connections with the malignant transformation, growth, proliferation, and metastasis of various cancers and solid tumors. Relevant connections between the PI3K/Akt/mTOR pathway, cell survival, and prostate cancer (PC) provide a great therapeutic target for PC prevention or treatment. Recent studies have focused on small-molecule mTOR inhibitors or their usage in coordination with other therapeutics for PC treatment that are currently undergoing clinical testing. In this study, the function of the PI3K/Akt/mTOR pathway, the consequence of its dysregulation, and the development of mTOR inhibitors, either as an individual substance or in combination with other agents, and their clinical implications are discussed. The rationale for targeting the PI3K/Akt/mTOR pathway, and specifically the application and potential utility of natural agents involved in PC treatment is described. In addition to the small-molecule mTOR inhibitors, there are evidence that several natural agents are able to target the PI3K/Akt/mTOR pathway in prostatic neoplasms. These natural mTOR inhibitors can interfere with the PI3K/Akt/mTOR pathway through multiple mechanisms; however, inhibition of Akt and suppression of mTOR 1 activity are two major therapeutic approaches. Combination therapy improves the efficacy of these inhibitors to either suppress the PC progression or circumvent the resistance by cancer cells.

## 1. Introduction

Prostate cancer (PC) represents the second highest form of cancer-related mortality in men [1]. The PC pathophysiology includes androgen-receptor signaling aberrations, deleterious somatic and germline abnormalities, irregularities of tumor suppressor and oncogenic genes, and genetic alterations [2]. Generally, PC can be categorized as a localized, primary castration-resistant PC (CRPC) form or as metastatic CRPC (mCRPC) forms. Although high long-term survival rates have been reported in localized PC, metastatic PC is quite challenging to cure, even after multimodal therapy [3]. The National Comprehensive Cancer Network (NCCN) proposed a categorization system for patients with PC as very low, low, intermediate, high, and very high risk groups [4]. The intermediate risk group has the largest number of patients, including two subsets; patient prognosis classified as favorable and unfavorable [5]. Current PC treatment strategies include expectant management (monitoring for PC progression without definitive therapy), surgery, and radiation for localized PC, while chemotherapies are the main treatment strategy for metastatic PC [6]. Androgen deprivation therapy (ADT) is the standard-of-care first-line chemotherapy for PC treatment [1]. It is known that inflammation and inflammatory mediators play a significant role in PC progression. Corticosteroids can act as endogenous cortisol or mineralocorticoids, thus inducing upregulation in the expression of anti-inflammatory proteins and downregulation in the expression of pro-inflammatory proteins through stimulation of the glucocorticoid receptors. These anti-inflammatory effects of corticosteroids may help to treat PC. In addition, corticosteroids may directly reduce tumor-induced pain, which is a secondary consequence of PC metastasis to other organs.

In clinical trials, the major treatment application is a combination of docetaxel-based regimen and corticosteroids (i.e., prednisone). Several other chemotherapeutic agents, including cabazitaxel, sipuleucel-T, and alpharadin, as well as the Food and Drug Administration (FDA)-approved androgen synthesis inhibitors abiraterone acetate (new-generation antiandrogens) and enzalutamide (the novel androgen receptor (AR) inhibitor), are regularly prescribed for PC treatment, depending on patient conditions and the extent and location of the disease [7]. While these therapies provide clinical benefits for PC, the survival rate of patients remains poor. Additionally, various adverse effects such as metabolic disorders, cognitive impairments, cardiac morbidity, and sexual dysfunctions have been reported. It has been proposed that chemotherapy is superior to ADT in terms of survivorship and metastatic PC resistance [6]. Current therapeutic approaches are searching for regimens that are capable of either generating sustained changes in tumor heterogeneity or agents that are able to interfere with the cellular and molecular pathways involved in PC initiation or progression, thus inducing genetic or functional modifications at the cellular level.

Alterations in multiple cellular signaling pathways are important hallmarks of carcinogenesis. Deregulation of pathways such as ARs, serine-threonine kinase (also known as Akt), nuclear factor-κB (NF-κB), Wnt, Hedgehog, and Notch, has been involved in both the initiation and progression of PC [8]. Additionally, data has shown that irregularity in the phosphatidylinositol 3-kinase (PI3K)/Akt/mTOR-signaling pathway has been implicated in various human cancer types including PC [9]. The PI3K/Akt/mTOR pathway is analogous with cell growth, development, proliferation, metastasis, malignant transformations, tumor progression, therapeutic resistance, and apoptosis [10]. Aberrant activity of PI3K/Akt/mTOR has been detected in multiple forms of human cancer due to the abnormal activation of both Akt and S6 or the phosphatase and tensin homolog (PTEN) suppression in cancer cell lines. Additionally, it has been found that the PI3K/Akt/mTOR-signaling is up-regulated in approximately 30–50% of PC subjects [11]. Changes in the molecular levels of PI3K/Akt/mTOR trigger several downstream targets, with their most significant role being in tumorigenesis. Given the therapeutic potential of natural-based preparations, these compounds are of great interest. To date, several natural bioactive compounds including afrocyclamin A, apigenin, arctigenin, curcumin, cryptotanshinone, oridonin, salidroside, and vitexin were reported to target the PI3K-Akt-mTOR pathway, although some compounds are currently under examination in clinical trials [12]. Development of mTOR inhibitors (rapamycin analogs) and improvement of their pharmacokinetic profiles via novel formulations facilitates future evaluation of their anticancer capabilities [13]. In this review, experimental and clinical evidence on the role of the PI3K/Akt/mTOR pathway in PC as well as the possible implications of treatment with natural or synthetic mTOR inhibitors have been discussed.

## 2. The mTOR Pathway

mTOR is a serine/threonine protein kinase associated with the phosphatidylinositol-3-kinase-related kinase (PIKK) family. mTOR is known to participate in cell development and regulation, autophagy, proliferation, survival, and autophagy. Additionally, mTOR has a key function in the proliferation, angiogenesis, and migration of cancer cells. [14]. mTOR is the catalytic core component of two complexes, namely mTORC1 and mTORC2. mTORC1 is a heterotrimeric protein kinase containing different mTOR catalytic subunits, including mTOR, PRAS40, DEPTOR, and tti1/tel225, as well as two associated proteins, namely the regulatory associated protein of mTOR (Raptor) and the mammalian lethal with sec-13 (mLST8). It has been shown that the activation of Akt stimulates mTOR kinase activity via the upregulation of the guanosine triphosphate (GTP) binding protein and Ras homolog enriched in the brain (Rheb). Thus, the mTOR pathway is largely regulated by the PI3K/Akt signals [15]. The components of mTORC1 include the core mTOR, mLST8, and Raptor subunits [16].

The mTORC2 complex includes mSin1, mLST8, DEPTOR, PROTOR1/2, tti1/tel2, and RICTOR. The tuberous sclerosis complex (TSC1/TSC2) is another downstream target of Akt, which can inhibit mTORC1 through GTPase activity of Rheb. TSC1/2 can regulate both mTORC1 and mTORC2 functions, while S6K1 only mediates mTORC2 [17]. mTORC1 participates in cell growth and metabolism, DNA damage, and hypoxia, while mTORC2 controls cell growth and proliferation through the PI3K pathway [16].

The abnormal regulation of mTOR has been extensively reported within human carcinomas from several different origins [14] and it is known that the PI3K–Akt pathway is the major upstream regulator of mTOR-signaling. Activation of mTORC, specifically mTORC1, induces the P70 S6 kinase (P70S6K) and the eukaryotic initiation factor 4E (eIF4E) binding protein 1 (4E-BP1) phosphorylation and activation, resulting in the elevation of the translation rate of several mRNAs that play a role in cell growth, metabolism, and oncogenic transformation [18].

## 3. Prostatic Neoplasms

PC is the second highest cancer-related cause of mortality in men [19]. The highest prevalence of PC has been reported in western countries and the lowest rate has been recorded in Asia. However, the number of PC is increasing in Asia due to westernization [20]. The majority of cancers are multifactorial, alike to PC. Current evidence suggests that aging increases the risk of PC, particularly after the age of 60 [21]. Additionally, men with a family history of PC have a greater chance of developing PC [22]. There is a significant interconnection between dietary patterns and the potential risk of PC, including the excessive consumption of red meat and dairy [23]. Vitamin D3 could inhibit PC cell growth and its deficiency provides an additional risk factor for PC [24]. It has been suggested that men with diabetes have a lower risk of PC [21].

ADT is the standard treatment for PC [25], although the disease can possibly progress into CRPC, which is lethal [26]. Radical prostatectomy and radiation therapy are aggressive local therapies for PC [27]. Current medications for PC include androgen inhibitors such as docetaxel, mitoxantrone, and prednisone. Docetaxel is a taxane that inhibits mitosis and the AR signaling. Cabazitaxel is another taxane which is specifically designed to overcome docetaxel-resistance [26,28]. Abiraterone inhibits androgenic steroid synthesis [29], while enzalutamide and darolutamide are AR antagonists [30]. Apalutamide, flutamide, and nilutamide are other common AR inhibitors [29,31]. Gonadotropin-releasing hormone (GnRH) agonists or antagonist therapy is a specific type of ADT. Goserelin and leuprolide acetate are GnRH agonists, whereas degarelix is a GnRH antagonist [32,33]. Abiraterone and enzalutamide are both FDA-approved for patients with CRPC [2,34]. Radium-223 was targeted for bone metastasis but further research demonstrated that patients with CRPC had increased survival after Radium-223 therapy [35]. A number of treatments have been developed in non-androgen-dependent manners such as sipuleucel-T, also known as an autologous form of cellular immunotherapy [36].

The serine-threonine kinase Akt from the mTORC1 complex amplifies the PI3K/Akt axis to stimulate cell growth by enhancing ribosome biogenesis and mRNA translations, and by increasing the protein expression, which together leads to the promotion of lipid synthesis and autophagy inhibition. Akt regulates the TSC2 phosphorylation at serine-939 and threonine-1462 by 5′ AMP-activated protein kinase (AMPK) [37]. TSC2 restricts the IKK/NF-κB activity through inhibition of mTORC1 in PTEN-null PC. In contrast, when TSC2 is mutated, it can promote IKK/NF-κB activity, which is an upstream of Akt and mTORC1 [38]. PTEN and P53 deletions or mutations were shown to act as regulators in CRPC. Loss of PTEN, alone or along with p53, causes hexokinase 2 (HK2) upregulation in PC cells. HK2 mediates aerobic glycolysis, which is necessary for tumor growth. Additionally, deletion of PTEN leads to an increase of HK2 protein levels through the Akt/mTORC1/4EBP1 signaling pathway [39] (Figure 1).

## 4. The PI3K/Akt/mTOR Signaling

As mentioned earlier, the PI3K pathway is significantly upregulated in PC cells. Additionally, PI3K activation is associated with Akt protein kinases, which has been implicated in PC progression in vivo. Activation of Akt induces the phosphorylation of TSC, which regulates the mTOR-signaling pathway. It has been shown that the tumor suppressor PTEN, located on chromosome 10, is the primary PI3K/Akt/mTOR negative regulator, which directly inhibits the action of PI3K [40]. PTEN is implicated in the modulation of cellular proliferation. PTEN deletion is the major cause of the PI3K pathway induction, resulting in an escalation in cell cycle activity and in the proliferation in human cancers such as PC. Akt has also a significant effect on cell survival through the Bcl-2-associated death (BAD) promoter, whereas mTOR is involved in cellular differentiation [41]. The downstream targets and major regulators of the translation of mTORC1 are 4EBP1 and S6K1, which are implicated in the regulation of cell proliferation. Overall, the PI3K/Akt/mTOR signaling actively participates in cellular homeostasis, indicating its possible therapeutic value for PC management [15,42].

## 5. Prostatic Neoplasms and PI3K/Akt/mTOR-Signaling

In prostate epithelial cells, overexpression of Akt or suppression of PTEN activity and PTEN loss-of-function results in PI3K/Akt/mTOR activation and is sufficient for the development of PC in vivo (Figure 1). In a study on the PTEN and RICTOR-deleted mouse model of PC, a blockade of mTOR inhibited prostate tumorigenesis in epithelial cells [43]. BEZ235, a dual inhibitor of PI3K and mTOR, reduced the tumor volume in a mouse model of PC, which was mediated by PTEN loss [44]. In another murine PC model, administration of various mTOR inhibitors such as rapamycin and everolimus led to antitumor action [44]. It was also demonstrated that the second generation of mTOR inhibitors such as AZD2014, AZD8055, CC-223, MLN0128, OSI-027, palomid 529, and torin-1/2 can directly target the ATP binding sites on mTOR and have the potential to impede the mTOR kinase activity, representing a profound antitumor efficacy in preclinical studies [45]. Therefore, various in vitro and in vivo studies were designed and targeted the PI3K/Akt/mTOR-signaling pathway to modulate the PC progression [46].

Leucine is an important dietary amino acid in humans that has the ability to stimulate protein biosynthesis through the phosphorylation of mTOR [47]. Prolonged consumption of a plant-based diet may lead to a decrease in levels of essential amino acids and insulin sensitivity. A decrease in serum levels of the insulin growth factor (IGF)-I results in an elevated risk of developing PC [48]. Accordingly, reduced IGF-I is thought to be a significant risk factor for PC, though it can suppress the mTORC1 activity caused by restriction of leucine [49].

ANXA7 is a Ca^+2^- and phospholipid-binding protein [50]. It was shown that (S)-ethyl1-(3-(4-chlorophenoxy)-2-hydroxypropyl)-3-(4-methoxyphenyl)-1H-pyrazole-5-carboxylate (SEC) can activate ANXA7 [51] while preventing the growth of PC cells through a blockade of the expression of several prometastatic genes (i.e., CCL2, APLN, and IL6ST) and the AMPK/mTORC1/STAT3 signaling pathway. In addition, the STAT3 activation has been attributed to the activation of mTORC1, while AMPK suppresses mTORC1 [52].

CCL2, also known as monocyte chemoattractant protein-1 (MCP-1), is a protein from the CC chemokine family [53] that has a low expression in PC cell lines. CCL2 can prevent PC cells from programmed cell death by increasing levels of survivin and levels of phospho-p70S6K in rapamycin-induced cell death [54], indicating the involvement of the mTOR pathway in PC [54]. CCL2 can increase the amount of mTORC1, unlike AMPK [55].

FGF23 is synthesized in bone osteocytes and osteoblasts and acts as a hormone to mediate the metabolism of phosphate and vitamin D. FGF23 can function as an autocrine, systemic, and a paracrine growth factor for PC, of which the production of autocrine is the most essential factor for PC progression [56]. FGF23 increased the anchorage-independent growth, invasiveness, and proliferation through the Akt and ERK signaling in PC cell lines [57]. It has been demonstrated that a reduction of iron levels activates the hypoxia-inducible factor-1 (HIF-1), thereby facilitating the transcription of FGF23 [58]. High dietary intakes of calcium and phosphorus may induce the risk of PC metastasis by increasing the systemic and paracrine production of FGF23. Overactivity of HIF-1 increased the autocrine FGF23 synthesis in PC [56].

Several in vitro and in vivo experiments suggest that in PC cells, the expression of the Golgi membrane protein 1 (GOLM1) is upregulated. GOLM1 is an oncogene that promotes the migration and invasion of cancer cells, and inhibits programmed cell death through stimulation of the PI3K/Akt/mTOR-signaling pathway, which may be a favorable option for the treatment of PC [59].

## 6. Preclinical Studies on Synthetic Agents Targeting mTOR-Signaling in Prostatic Neoplasms

Many synthetic agents have been introduced to possess protective or therapeutic properties in conditions of both acute and chronic intestinal inflammation through several mechanisms including the inhibition of oxidative stress and inflammatory pathways or by preservation of the integrity and functionality of intestinal wall tissue. Many studies reported that the mTOR-signaling pathway was a target of these synthetic products against prostatic neoplasms. Thereby, we present data that confirm mTOR is a major and novel regulator of inflammation in the treatment of prostatic neoplasms. In the next sections, a number of synthetic mTOR inhibitors (direct or indirect) that interfere in PC treatment via the mTOR pathway will be discussed (Table 1 and Table 2).

### 6.1. Everolimus

Everolimus is a mTORC1 inhibitor. The drug is approved by the FDA for the treatment of adults with progressive, well-differentiated, and non-functional neuroendocrine tumors of gastrointestinal or lung origin, with unresectable, locally advanced, or metastatic disease [60]. Everolimus binds to the FK506 binding protein 12 receptor, which prevents mTORC1 activation [18]. Everolimus also enhances apoptosis and decreases the SK1 and vascular endothelial growth factor (VEGF) expressions that are involved in P70S6K phosphorylation. Docetaxel upregulated the VEGF mRNA level, but unlike everolimus, it had no effect on VEGF secretion. Their combination downregulated the p-P70S6K, SK1, VEGF, and CD31 expressions in breast cancer and hormone-insensitive PC [61].

**Table 1 pharmaceutics-13-01195-t001:** In vitro studies on synthetic PI3K/Akt/mTOR inhibitors in PC.

Treatment	Cell Line Name	Mechanism of Action	Results	Reference
Everolimus (0.5–5 nM) and Torkinib (PP242) (0.5–5 nM)	DU145 & 22RV1 PC	Everolimus: mTORC1 inhibition PP242: dual mTORC1/mTORC2 inhibition	↓cell growth ↓proliferation rates	[62]
Rapamycin (10 nM)	PC3, MDCK, and COS-7	mTORC1 inhibition	Prostate tumor overexpressed -1, stimulates protein translation in an mTORC1-dependent manner	[63]
Rapamycin (250 nM); Torin-1 (250 nM); and Cycloheximide (10 µg/mL)	MEFs (WT(4EBP1/2+/+; p53−/−) and DKO (4EBP1/2−/−; p53−/−)	Torin-1: ATP-competitive inhibitors of mTORC1 Torin-1 blocked canonical mTORC1-dependent events (phosphorylation of S6K1 and 4E-BP1), ↓35S-Cys/Met incorporation, ↓translation of eIF4B, and ↓cytoplasmic ribosomal proteins Rapamycin: ↓mTORC1 kinase activity	4E-BPs mediate acute mTOR inhibition ↓translation of TOP and TOP-like mRNAs TOP mRNAs require eIF4G1 to anchor eIF4E to the cap →translational regulation by the 4E-BPs and mTORC1	[64]
Rapamycin (0.1–1000 nM) and p-XSC (0.625–10 μM)	LNCaP, C4-2, and DU145	↓p-XSC →Akt expression (by inhibition of mTORC2) Rapamycin: ↓mTORC1 kinase activity	↓cell growth and viability ↓phosphorylation of the mTORC2 and mTORC1 downstream targets, Akt, RPS6, phospho-RPS6 (Ser235/236), and PKCα ↓IC50 values p-XSC→↓levels of Rictor, Combination more effectively inhibited viability and mTOR signaling	[65]
Rapamycin (20 ng/mL)	PC3	Inhibition of mTORC1	↓translation in normoxic cells and 2-2-under hypoxic conditions	[66]
Rapamycin (100 nM)	PC3 and LNCaP PC	Rapamycin: inhibitor of mTOR activity	↓NF-κB DNA-binding activity ↓IKK activity ↓RelA/p65phosphorylation→ blocked ↓S6K phosphorylation	[67]
LY294002 (10 nM) and Rapamycin (100 nM)	PC3 and LNCaP	LY294002:PI3-kinase antagonist Rapamycin: inhibitor of mTOR activity	↓NF-κB-dependent reporter activity ↓RelA/p65phosphorylation	[67]
Bicalutamide (10 µM–50 μM) and Ridaforolimus (0.5 Nm–50 nM)	LNCaP and C4-2	Ridaforolimus: ↓mTOR signaling, ↑p-Akt levels, and ↓phospho-S6 levels Bicalutamide: AR blockade	↑cell cycle arrest ↓S and G2/M phases and G1 arrest Ridaforolimus→↑PSA expression Bicalutamide→↓Ridaforolimus-induced stimulation of PSA ↓cell proliferation	[68]
Ridaforolimus (0.0001–1000 nM)	LNCaP and C4 (PTEN−/−)	Ridaforolimus: ↓mTOR signaling, ↑p-Akt levels, ↓phospho-S6 levels	↓cellular proliferation	[68]
Bortezomib (10, 100 nM) (in hypoxic or normoxic condition)	LNCaP and PC3	↓PI3K/Akt/mTOR and MAPK pathways ↓p44/42 MAPK phosphorylation	↓HIF-1α levels transcriptional activity and protein expression ↓phosphorylation of the downstream targets of Akt, p70S6K^(Thr389)^, and S6RP^(Ser235/236)^ (hypoxic condition) ↓nuclear HIF-1α levels ↑ cytoplasmic HIF-1α (hypoxic condition) ↓baseline VEGF secretion levels ↓proteasome activity protein levels of total Akt26 and total amount of p44/42 MAPK →no change	[69]
Rapamycin (20 nM) and Torin-1 [(PC3 & DU145 (250 nM), LNCaP (125 nM)]	DU145, LNCaP, and PC3	Torin-1→↓ phosphorylation of pro-ADM1	AMD1 mRNA expression: In PC3 with Torin, was less than Rapamycin. In LNCaP with Rapamycin, was more than Torin. In DU145 with Rapamycin, was more than Torin.	[70]
Torin-1 (250 nM) and Torin-1 (250 nM) + MG132 (5 µM)	DU145 cells expressing Myc-AMD1-HA	Torin-1→ mTORC1 inhibition→↓proAMD1 stability	↓half-life of proAMD1	[70]
Afrocyclamin A (50 pM); Wortmannin (30 nM) + afrocyclamin A; LY294002 (20 mM) + afrocyclamin A; Rapamycin (100 nM) + afrocyclamin A; and Actinomycin D (5 mg/mL) + afrocyclamin A	LN1	Rapamycin and Actinomycin D: ↓p70S6K→↓mTORC1-Akt pathway signaling LY294002 and Wortmannin: inhibition of PI3K→↓p-Akt→↓PI3K/Akt/mTOR signaling	↓pS6-Kinase and 4EBP1 ↓PSA synthesis and cell proliferation	[71]
Venuloside A (50 μM)	LNCaP-FGC and PC3	ESK242 and ESK246: ↓p70S6K→↓mTORC1 activity	↓Leucine uptake ↓cell viability in LNCaP and PC3 (only ESK242) downregulation of CDK1 and UBE2C	[72]
Salinomycin (4 µM)	PC3 and DU145	↓p70S6K & pS6→↓activation of mTORC1	↓LRP6 expression and phosphorylation ↓axin2 expression ↓Wnt/βcatenin signaling ↓ colony formation ↓ cell growth ↓cancer cells viability	[42]
Dactolisib (0.5–1.5 µM) and Rapamycin (100 nM)	UMN and 4240P	BEZ235: ↓activation of 4EBP1 and 70S6K→↓mTORC1-Akt pathway signaling Rapamycin: ↓only p70S6K→↓mTORC1-Akt pathway signaling	↓HK2 protein expression	[25]
Sapanisertib (100 nM)	PC3 and LNCaP	Blocked of mTORC 1/2 and downregulation of mTOR regulator genes→ ↓mTORC1/2 activation	Downregulation of cyclin D1 and HIF 1α/2α ↑apoptosis no effect on androgen receptor	[34]
3- carboxymethylpyrrolidine -2,4-dicarboxylic acid (2 µmol/L)	PC3 and DU145	↓ phosphorylation of NF-κB and IκBα ↑cellular caspase-3/7 activity ↓colony formation ↑cell apoptosis ↓expression of survivin ↑expression of cleaved caspase-3	↓cancer cells growth ↓proliferation	[73]
LY294002 (10 μM)	PC3	↓phosphorylation of Akt→↓mTOR activity	↓pTSC2, pS6K, and p65 No effects on expression of pr regulation of NF-κB through mTORC1	[24]
Rapamycin (100 nM) + IKKβ inhibitor (5 μM)	TSC2 null	Rapamycin: ↓p70S6K→↓mTORC1-Akt pathway signaling	↓mTORC1 activity ↑NF-κB phosphorylation ↓phosphorylation of mTORC1 ↓IRS1	[24]
Rapamycin (1 nM) and Bicalutamide (10 μM)	LNCaP and C4-2	Expression of Akt inhibited apoptosis induced by combination of rapamycin and Bicalutamide mediated by mTORC1	Induction of apoptosis in C4-2 cells by inhibition of both Raptor and Rictor Bicalutamide→ ↑Raptor inhibition Overexpression of both Raptor and Rictor → inhibition of apoptosis induced by Rapamycin and Bicalutamide mTORC1 inhibited while mTORC2 stimulated AR activity and Akt phosphorylation ↑AR transcriptional activity in Rapamycin-treated cells by active form of Akt (pCMV-6-myr-Akt-HA) mTORC1 and AR activate parallel cell survival pathways	[74]
Flutamide (40–70 µg/mL) + temsirolimus (CCI-779) (0.01–800 nM)	PTEN^−/−^TP53^−/−^ stem/progenitor prostate epithelial cells	mTOR and AR signaling are required clonogenic and have tumor-initiating activity in PTEN^−/−^TP53^−/−^ prostate progenitor cells	↓tumor growth and size	[75]
8-CPT-2Me-cAMP (100, 150 µM); LY294004 (20 µM); and Rapamycin (100 µM)	1-LN	Expression of Akt and mTOR inhibited	CPT-2Me-cAMP (100 µM): ↑ levels of Raptor and Rictor ↑ levels of p-PRAS40 and GbL induction of Epac1 signaling →activation of mTORC1 and mTORC2 CPT-2Me-cAMP (150 µM): upregulation of p-AktT308 and p-AktS437 expression ↑sensitivity to LY294002 LY294002: suppression of 8-CPT-2MecAMP- dependent p-AktS473 and p-AktS473 kinase activities	[76]
Salinomycin (50 nM)	AR-expressing LNCaP (castration-sensitive) and C4-2B (castration resistant) PC cells	↓phosphorylation of RPS→↓activity of mTORC1	↓cytostasis ↓apoptosis ↓ autophagy ↓AR mRNA and protein levels	[37]
Fingolimod (a derivative of FTY720 or SPS-7) (10 µM)	LNCaP and DU-145	↓phosphorylation of p70S6K, Akt and 4EBP1→↓Akt/mTOR signaling pathway	↓apoptosis ↓cell growth ↓ cell cycle in G1	[77]
Fenofibrate (50 μM)	PC3	↓phosphorylation of p70S6K→↓phosphorylation of mTOR	↓aroliferation ↓apoptosis ↑AMPK and MAPK phosphorylation ↓cell growth	[78]
CORM-2 (40 µM)	LNCaP and PCa	Expression of LKB1→ ↑AMP/AMT radio→↓mTOR	↓apoptosis ↓proliferation ↑active form of caspase	[79]
mTOR shRNA-expressing lentivirus (LV-shmTOR) (20–40 μg)+ vector-derived LV as control (LV-shCON) (20–40 μg)	RWPE1, LNCap, and C4-2b	↓Expression of Akt and mTOR	↓proliferation in prostate cancer cells in mTOR shRNA-transduced cells ↓Akt, PI3K, S6K, and 4EBP1 (inC4-2b cells)	[80]
Panobinostat (10 nM) + dactolisib (BEZ235) (500 nM)	PC3 and PC3-AR	BEZ235-induced inhibition of downstream targets of mTORC1, p-4EBP1, and p-S6K ↓p-Akt, p-ATM & ↑apoptosis	Upregulation of ATM-Akt-Erk1/2 signaling → antitumor activity	[81]
VS-5584 (SB2343) (4–1000 nM)	PC3 (with PTEN deletion) MV4-11 cells (FLT3-ITD)	In PC3 cells: inhibition of both PI3K and mTOR-signaling cascade In MV4-11 cells: blocked of pAkt(S473) and pAkt (T308)	The enzyme inhibitory properties of VS-5584 translates into the modulation of the PI3K/mTOR-signaling pathway	[82]
Nitroxoline (1–30 μM)	PC3 and LNCaP	AMPK phosphorylation → activity of TSC1/TSC2 complex → mTOR inhibition →↓cell proliferation LC3-II protein →cell apoptosis	↑AMPKα phosphorylation ↓mTOR phosphorylation ↑LC3-II protein	[83]
NSK-01105 (a sorafenib derivative) (10 μmol/L) + sorafenib (10 μmol/L)	LNCaP and PC3	↓EGFR activity→ blocking the Raf/MEK/ERK and Akt/mTOR pathway	↓Raf-1 kinase activity ↓EGFR phosphorylation by NSK-01105 ↓MEK and ERK phosphorylation in both treatments ↓phosphorylation of Akt by NSK-01105	[84]
Voxtalisib (XL765) (5 μM) + pilaralisib (XL147) (10 μM) + rapamycin (1 µM)	PC3, LNCaP, C-81, C4-2B, 22rv1, DU145, and everolimus-resistant PC3 cells	PI3K/Akt pathway→ cell proliferation AKT/mTORC1/4EBP1 signaling pathway→ tumor cell growth	↓phosphorylation of Akt and PDK1 ↓4EBP1 phosphorylation ↓cell proliferation ↑ apoptosis	[85]
Diaminobenzidine (1.7–35.8 ng/mL)	PC3, DU145, LNCaP, and PNT1A	↓ amounts of RICTOR and RAPTORs ↓ mTOR activity	↓phosphorylation of Akt and 4E-BP1 No change in pS6K and pSG upregulation of mLST8	[34]
Salinomycin (50–400 nM)	RWPE-1, LNCaP, and C4-2	Salinomycin →inhibiting AR Salinomycin →inhibiting the PI3K/Akt/mTORC1 pathway	↓cell proliferation ↑sensitivity Salinomycin (200 nM): ↓RWPE-1 Salinomycin (200 nM): ↓castration-resistant in C4-2 400 nM Salinomycin: ↓RWPE-1 less than C4-2 ↑autophagy Robust cleavage of PARP1 and procaspase-3 in C4-2 ↓phosphorylation of mTOR ↓phospho-AR in LNCaP	[37]
Remotiflori radix (100–500 mg/mL)	PC3 and DU145	↑ cells in G1, ↓ cells in S, and G2/M→ cell death	↑ cell death	[86]
Remotiflori radix (250, 500 mg/mL)	PC3, DU145, and HT1080	LC3 expression → ↓ autophagy YO-PRO-1 uptake → early cell apoptosis AMPK phosphorylation → ULK1 protein kinase activation→ ↓ mTORC1 activity → cell death	↓LC3 expression ↑YO-PRO-1 uptake ↑cleaved caspases and PARP cleavage ↑AMPK phosphorylation ↓mTORC1 activity	[86]
Metformin (1585 μg/mL); Vitamin D3 (400 μg/mL); and Metformin (1585 μg/mL) + Vitamin D3(400 μg/mL)	DU145	G1/S cell cycle arrest → cell death ↑ AMPK phosphorylation → inhibition of the mTOR/S6K signaling pathway → ↓ cell proliferation	↓proliferation Combination of Metformin (1585 μg/mL) and vitamin D3 (400 μg/mL) Reduced proliferation rate	[24]
CCL2 (100 ng/mL)	PC3	CCL2 treatment→AMPK phosphorylation→mTORC1 signaling inhibition and autophagy induction	Induction of mTORC1 activation and downregulation of AMPK/Raptor phosphorylation in serum-starved PC3 cells → cell survival AMPK directly phosphorylates Raptor on Ser^792^ to inhibit mTORC1 D942(AMPK activator) induces Raptor phosphorylation and downregulates mTORC1 signaling, promoting cell death in PC3 but CCL2 reverses the lethal effect of D942	[55]
Rapamycin (1 µM)	SH-SY5Y	Down-regulation of PINK1 by siRNA and treatment with rapamycin reduced the number of migrated cells, and if PINK1 activates mTORC2, overexpression of PINK1 may affect cell motility	PINK1 induces phosphorylation of Rictor → activation of mTORC2 → stimulation of cell motility	[87]
Docetaxel (5 nM); Everolimus (0.5–5 nM); and Everolimus (0.5–5 nM) + docetaxel (5 nM)	PC3	mTOR induction→↑ SK1 expression→↑ VEGF→↑ P70S6K phosphorylation	Docetaxel: no change in P70S6K phosphorylation and SK1 mRNA↑ VEGF mRNA Everolimus alone or with docetaxel: ↓P70S6K phosphorylation, ↓SK1 mRNA,↓ VEGF expression and secretion	[61]
7-aza-tetrahydroquinazoline (140 nm)	NCI-PC3 (PTEN-null)	Inhibit both mTORC1 and mTORC2	Compound 12 h→the most potent compounds 12a-h→ potency and selectivity for mTOR over PI3Kα and PI3Kδ	[88]
7-aza-tetrahydroquinazoline (140 nm)	NCI-PC3 (PTEN-null)	Inhibit both mTORC1 and mTORC2	Selectivity for PI3Kα↑ ↑interaction with Trp2239 in mTOR ↑antiproliferative	[88]

Abbreviations: ↓ = decrease; ↑ = increase; → = leads to; AMD1 = adenosylmethionine decarboxylase-1; AMPK = 5′-AMP-activated protein kinase; CCL2 = (C-C motif) ligand 2; CORM-2 = carbon monoxide–releasing molecule-2; 8-CPT-Me-cAMP = 8-(4-chlorophenylthio)-2′-*O*-methyladenosine 3′,5′-cyclic monophosphate; IKKβ = IκB kinase β; LY294002 = 2-(morpholin-4-yl)-8-phenyl-4H-1-benzopyran-4-one; MEKK3 = mitogen-activated protein kinase kinase kinase 3; MG132 = carbobenzoxy-Leu-Leu-leucinal; PI3K = phosphatidylinositol 3-kinase; PINK1 = PTEN-induced kinase 1; p-XSC = p-xyleneselenocyanate; SK1 = sphingosine-kinase-1; TOP = terminal oligopyrimidine; and VEGF = vascular endothelial growth factor.

**Table 2 pharmaceutics-13-01195-t002:** In vivo studies on synthetic PI3K/Akt/mTOR inhibitors in prostate cancer.

Treatment	Animals	Model	Number of Animals	Treatment Duration	Results	Reference
Sapanisertib (1 mg/kg/day, oral)	Nude mice	PC3 tumor xenograft	*n* = 10	21 days	↓PC3 xenograft growth ↓tumors weight ↑apoptosis Inactivation of mTORC 1/2 Downregulation of cyclin D1 and HIF1α	[34]
Palomid 529 (100 mg/kg, s.c.)	Male CD1 athymic nude mice	PC3 or 22rv1 tumor xenograft	*n* = 48	4 weeks	↑radiation effect ↓growth rate, tumor volume, and tumor weight ↑apoptotic cells→↓ proliferation index and the number of vessels	[89]
Bicalutamide (10 mg/kg, i.p.); Ridaforolimus (0.3 mg/kg, i.p.); and Bicalutamide (10 mg/kg, i.p.) + ridaforolimus (0.3 mg/kg, i.p.)	Male nude mice	C4-2 tumor xenograft	*n* = 10	21 days	↓tumor growth in single agent and combination Combination →↑ antitumor activity Combination →↓ plasma PSA level	[68]
Everolimus (7.5 μg/g, i.p.)	Aged *Tsc1 KO* mice	Mouse model of TSC1 deletion in prostate epithelium	*n* = 6	3 times/per week (for 4 weeks)	Inhibition of the mTORC1 pathway in the testis and prostate everolimus → antiproliferative effect in testis	[90]
Everolimus (10 mg/kg/day, oral)	Mice	Prostate hyperplasia in PB-Rheb transgenic mice	*n* = 6	Daily for 1 week	Akt Rheb overexpression → relief of Akt inhibition by PTEN haplo-insufficiency and↑ of mTORC1 level	[91]
Styrene sulfonamide acid (2–10 mg/kg, i.p.) and Fingolimod (FTY720) (5 mg/kg, i.p.)	Nude mice	PC3 tumor xenograft	*n* = 6	14 days	↓apoptos ↓cell growth ↓tumor growth ↓mTOR/Akt pathway	[77]
NSK-01105 (a sorafenib derivative) (7.5–30 mg/kg/day, oral) and Sorafenib (30 mg/kg/day, oral)	Male BALB/c nu/nu nude mice	LNCaP and PC3 tumor xenograft	Sorafenib group in LNCaP model: (*n* = 4) Other groups: *n* = 6	21 days	↑inhibition rate in both models ↓phosphorylation of ERK and mTOR ↓Akt phosphorylation ↓Bcl-2	[84]
IGFBP-3 (5 × 10^8^ PFU in PBS, i.p.) and IGFBP-3 + IL-24 (5×10^8^ PFU in PBS, i.p.)	Athymic nude mice	LNCaP tumor xenograft	*n* = 6	Every day (24 days)	tumor size ↑expression of PARP ↓mTOR	[92]
Salinomycin (5 mg/kg; C4-2 cells: oral gavage; and LNCaP-II cells: i.p.)	Nude male mice	LNCaP-II and C4-2 tumor xenograft	Control: *n* = 5 Salinomycin: *n* = 3 LNCaP-II: *n* = 5 C4-2: *n* = 4	LNCaP-II: every 3^rd^ day until day 16 C4-2: every 2^nd^ day until day 21	Salinomycin in LNCaP-II:↓ CYP17A1 and P-RPS, and ↓ tumor size Salinomycin in C4-2: ↓tumor size and ↓phospho- TSC2	[37]
Salinomycin (5 mg/kg, oral)	Nude male mice PTEN^pc−/−^	PC3 and LNCAP tumor xenograft	*n* = 2	21 days	↓cytostasis ↓apoptosis ↓autophagy ↓mTORC1 activity ↓tumor growth	[37]
Everolimus (10 mg/kg, i.p.)	Mice	Xenograft prostate tumor (injection of AMD1 silencing or AMD1 ectopic expression cells)	*n* = 3	-	↓AMD1 pro expression level	[70]
Docetaxel (5 mg/kg, i.p.); Everolimus (5 mg/kg, i.p.); and Docetaxel (5 mg/kg, i.p.) + everolimus (5 mg/kg, i.p.)	BALB/c nude male mice	PC3 tumor xenograft	*n* = 4	Twice a week (3 weeks)	↓tumor volume with RAD + Doc	[61]
Apitolisib (GDC-0980) + P-GDC-0980 (5 mg/kg, i.v.) and Docetaxel + P-docetaxel (10 mg/kg, i.v)	Nude mice	PC tumor xenograft	*n* = 4	GDC-0980 + P-GDC-0980: twice per week and Docetaxel + P-docetaxel: one single dose	↑antitumor effect	[93]

Abbreviations: ↓ = decrease; ↑ = increase; → = leads to; AMD1 = S-adenosylmethionine decarboxylase 1; ANXA7 = annexin A7; elF4G = eukaryotic translation initiation factor 4G; ERK = extracellular signal-regulated kinases; IGFBP-3 = insulin-like growth factor binding protein-3; IL-24 = interleukine-24; LC3 = light chain 3; P-docetaxel = polymer-docetaxel; P-GDC-0980 = polymer GDC-0980; PBS = phosphate-buffered saline; PTEN = phosphatase and tensin homolog; s.c. = subcutaneous; and TSC2 = tuberous sclerosis complex 2.

### 6.2. Rapamycin

Various in vitro studies reported that rapamycin inhibited both the cellular mTORC1 and mTORC2 pathways in several cancers such as renal, multiple myeloma, leukemia, and lymphoma [94]. Data indicated that rapamycin prevented the ISCs progression during caloric restriction via the modulation of the mTORC1 pathway. Rapamycin can induce autophagy; thus, it may have a beneficial impact on autophagy related diseases. Rapamycin binds FKBP12, which can suppress mTORC1 but not mTORC2 [95]. Recent studies proposed that rapamycin/FKBP12 inhibited de novo production of mTORC2 [96]. Although rapamycin inhibited mTORC1/2, long-term treatment caused resistance and was not suitable for monotherapy. A combination of rapamycin and bicalutamide (anti-androgenic drug) improved anti-prostate cancer effect due to the suppression of mTORC1 stimulated AR transcriptional activity [74]. Rapamycin prevented the 4EBP1 phosphorylation through induction of the Akt/mTORC1/4EBP1 signaling pathway [85] (Table 1).

### 6.3. Ridaforolimus

Ridaforolimus (deforolimus; AP23573; MK-8669) is a novel sirolimus derivative. It is a small-molecule kinase inhibitor of the mTOR and currently is in clinical development for the treatment of PC. Both intravenous and oral formulations of the agent are being tested in cancer clinical trials. In preclinical and clinical studies, ridaforolimus exhibited significant antitumor activity with acceptable safety and tolerability in PC patients. Recently, a phase III study demonstrated an improvement in progression-free survival when patients with at least a stable disease after treatment with standard chemotherapy received ridaforolimus compared to the placebo. Overall, these findings show the broad inhibitory effects of ridaforolimus on cell growth, cell division, metabolism, and angiogenesis, and support the use of intermittent dosing as a means to optimize antitumor activity, while minimizing side effects [97,98].

### 6.4. Salinomycine

Salinomycine affects the mTORC1 and Wnt/β-catenin signaling pathways by suppressing the expression of LRP6 [99]. It was shown that salinomycin significantly blocked the p70S6K and S6 activation in cancer cells, resulting in mTORC1 and cell development inhibition through the induction of apoptosis. Additionally, inhibition of mTORC1 led to the suppression of AR mRNA expression and protein level, illustrating the inhibitory effect of salinomycin on crosstalk between these pathways. In C4-2 cells, salinomycine reduced the AMPK expression, resulting in inactivation of mTORC1 [37].

### 6.5. Sipuleucel-T

Sipuleucel-T (STN: BL 125197) is a FDA-approved form of autologous cellular immunotherapy which is used in asymptomatic or minimally symptomatic mCRP cancer treatment methods [36]. Its purpose is to block the immune response to prostatic acid phosphatase expressed by cancer cells and to promote the activity of T cells [100]. While sipuleucel-T improved the immune system, the survival rate was low [101]. The outcomes of critical immunotherapy for prostate adenocarcinoma treatment (IMPACT) in a phase III study exhibited a significant improvement of 4.1 months in the median overall survival in the treatment group compared to the placebo group. There was no significant difference in median time to objective disease progression between the two patient cohorts [102].

### 6.6. Temsirolimus

Temsirolimus is a mTOR inhibitor used for the treatment of various forms of cancer. Temsirolimus arrested the cell cycle at G1/S in PC cells, thereby inhibiting tumor proliferation and angiogenesis by decreasing VEGF. The combination of temsirolimus and chemotherapy has a stronger anticancer effect than using each one alone [103].

### 6.7. Tetrahydroquinolines

Tetrahydroquinolines are a group of low-potency compounds with PC-inhibitory effects and have better selectivity for mTOR inhibition than 7-aza-tetrahydroquinazolines. These compounds have significant selectivity for mTOR compared to both PI3Kα and PI3Kδ, and are stable under typical conditions [88]. A combination of 7-aza-tetrahydroquinazolines and 2-S methyl morpholine can suppress both mTORC1 and mTORC2 [104], while reducing the p70S6K and Akt activities. Both compounds inhibited the mTORC2 complex by suppressing the Akt phosphorylation (serine 437) and the mTORC1 complex by inhibiting the S6 (serine 235 and 236) activity. They also activated the mTOR/Akt/PI3K pathway due to the loss of PTEN [88].

## 7. Clinical Implication of mTOR-Signaling Inhibitors in Prostatic Neoplasms

Although AR targeted therapies are proper solutions to treat mCRPC, their transient responses lead to a great interest on targeting the PI3K/Akt/mTOR signaling [105] (Table 3). Despite great improvement in PC diagnosis and the availability of drugs such as sipuleucel-T, successful treatment has not been achieved yet [106].

In a phase II study, temsirolimus was administered to men diagnosed with CRPC. Temsirolimus inhibited mTOR/TORC1, which was evidenced by monitoring circulating tumor cells (CTC), prostate-specific antigen (PSA) levels, progression-free survival (PFS), and overall survival (OS) times. Patients with primary tumor resistance, mutual AR activation, or both experienced an improvement of PSA and CTC. The combination of an antiandrogen and temsirolimus exhibited a rapid improvement in the response to treatment, in addition to a decrease in PSA, but this therapy was effective only for a short term [107]. In a single agent study, temsirolimus therapy resulted in stabilizing the disease and PSA reduction; however, overall median survival wasn’t satisfactory. Common adverse effects such as grade 3–4 thrombocytopenia and toxicity were also reported [108]. Moreover, concurrent therapy with VEGF targeting drugs such as bevacizumab showed a decline in CTC and PSA levels but the responses did not last for a long time and toxicities affected the intensity of dosage [108,109].

Apart from inhibition of AR and mTORC1, other possible pharmacological actions of salinomycin are unknown. In a phase II study, a combination of bicalutamide, everolimus, and salinomycin, as mTORC1 and AR inhibitors, could improve mCRPC by targeting the PI3K/Akt pathway [110,111]. In another phase I/II trial, a combination of everolimus and gefitinib [epidermal growth factor receptor (EGFR) inhibitors] caused a rapid increase in PSA level, though the standardized uptake value was transient, thus it was not able to affect tumor growth significantly. The most prevalent adverse effects were grade 2–3 of fatigue and hyperglycemia. There was no significant PSA reduction, while antitumor activity was observed [112].

In another phase II study, mCRPC patients were treated with everolimus, a TORC1 inhibitor. A rapid PSA progression was observed, although the PSA level dropped following a blockade of everolimus [113]. Single therapy with everolimus in patients with mCRPC was not effective on tumors or PSA, demonstrating that the modification of some mTOR mediators may be a reason for trivial clinical outcomes [114]. Another combination therapy including everolimus, carboplatin, and prednisone was slightly effective in PC treatment, though there were grade 3 and 4 toxicities [115].

Disease stabilization was observed in almost half of taxane-treated CRPC patients within a single agent phase II trial using ridaforlimus. The drug was well tolerated, though it did not reveal any significant outcome and increased the PSA level [97]. In another trial using ridaforlimus and bicalutamide as interventions, no significant effect was observed in PC improvement and even grade 3 adverse effects were recorded [98].

**Table 3 pharmaceutics-13-01195-t003:** Clinical studies on PI3K/Akt/mTOR inhibitors in prostate cancer.

Treatment	Study Design	Mechanisms and Effects	Adverse Effects	Duration	Reference
Temsirolimus (25 mg/weekly, oral) + bicalutamide, Flutamide, or nilutamide (at FDA-approved doses, oral)	Phase II trial, men with mCRPC (whom previous hormonal and first-line chemotherapy had failed), *n* = 20	↓ median CTC Short-termpartial Radiographicresponse Temsirolimus→ mTOR/TORC1 inhibition Combination therapy→ ↑ Response speed	Grade 3–4 of serious adverse events	12 weeks	[107]
Phase I: Temsirolimus (20 or 25 mg, oral) + bevacizumab (10 mg/kg, i.v.) Phase II: Temsirolimus (25 mg, oral) + bevacizumab (10 mg/kg, i.v.)	Phase I/II trial, men with mCRPC Phase I: *n* = 22 Phase II: *n* = 28	↑ PSA (32%) CTC levels↓ (82%) No disease progression (90% Cl)	Common ADEs: lymphopenia and thrombocytopenia	3 weeks	[109]
Sapanisertib (5 mg, oral) once daily	Phase II clinical trial, mCRPC patients pretreated with abiraterone acetate and/or enzalutamide, *n* = 9	↑PSA level (median = 159%, range: 12–620%), no decrease in CTC count, PSA levels, and phosphorylation of Akt and 4EBP1, ↓rpS6 phosphorylation (*n* = 2), ↑eIF4E activity (*n* = 2), poor inhibition of downstream signaling targets, no clinical response	Common ADEs: urinary frequency, pain, dyspnea, edema, mucositis, rash, and delirium	3–30 weeks (median time: 11 weeks)	[116]
Abiraterone acetate (1000 mg) + prednisone (5 mg twice daily, oral) + dactolisib (200 mg, twice daily, oral)	Phase I clinical trial 3 + 3 dose escalation design Progressive mCRPC, *n* = 6 (all discontinued)	No PSA decline or objective response were observed, the combination was poorly tolerated	Common ADEs: musculoskeletal pain, hypotension, dyspnea, and (grade 4) pneumonitis	27 days (range: 3–130 days)	[117]
Everolimus (5–10 mg, daily for 2 weeks, oral) Radiation therapy (5 days per week)	Phase I clinical trial, 3 + 3 dose escalation design, non-randomized, single-institution, open-label, PC with adenocarcinoma of the prostate following prostatectomy, ECOG 0–1, *n* = 18	Undetectable PSA (*n* = 9), sBCR development, ≤10 mg everolimus → safe and tolerable, remarkable anti-tumor responses CI:95%	Common ADEs: rash, urinary symptoms, and grade 3 acute toxicity (rash, anemia, lymphopenia, and neutropenia)	17.8 months (range 1.2 –46.0 months)	[118]
Bicalutamide (50 mg, oral) and Everolimus (10 mg, oral) (both once daily)	Phase II clinical trial single-arm, mCRPC, progressive and Bicalutamide-naive CRPC, pretreated with androgen deprivation, 83% hormone and/or chemotherapy-treated *n* = 24	↓PSA (*n* = 15), the median OS: 28 months, the median PFS:9.4 months, combination was effective in CRPC (95% Cl), significant numbers of everolimus-related toxicity	Grade 3 or 4 toxicities, neutropenia, anemia, thrombocytopenia, lymphopenia, pneumonia, pneumonitis, thromboembolic, abdominal pain, DVT/PE, neutropenia, right hip pain, renal failure, and non-neutropenic sepsis	28 months (14.1–42.7)	[119]
Carboplatin (10–20 mg/kg, oral) + Everolimus (10 mg, once daily, oral) + prednisone (5 mg, twice daily, oral)	Phase II clinical trial, fixed sample size, progressive mCRPC pretreated with docetaxel-based regimen, ECOG 0–2, *n* = 26	Median OS: 12.5 months, ↓ PSA > 30% (*n* = 4), median TTP: 2.5 months, and ↓ median CTC63%, no correlation of TSC1 gene mutations with clinical outcome, minimal efficacy in mCRPC, no objective responses	Grade 3 or 4 cytopenias, thrombocytopenia, and neutropenia	Median time: 2.5 months (1.8–4.3)	[115]
Phase I: Everolimus (30–70 mg, oral) on day 1 and/or gefitinib (250 mg/day, oral) Phase II: Everolimus (70 mg, oral) weekly + gefitinib (250 mg, oral) daily	Phase I/II clinical trial, single-agent (phase I), a Simon two-stage design (phase II), progressive metastatic PC, total number = 37 Phase I: *n* = 10 (5 patients discontinued) Phase II: *n* = 27 (18 patients discontinued)	↑tumor growth, expression of PSA↑, combination therapy→ no antitumor activity, ↓the early FDG SUV consistent with mTOR inhibition (*n* = 27) and PSA ↑ (*n* = 20), SD (*n* = 8) 95% CI	Hyperglycemia, thrombocytopenia (grade 4), and renal failure lymphopenia (grade 3)	Median time: 12 weeks→ phase I 11 weeks → phase II	[112]
Temsirolimus (25 mg, oral) per week (i.v.) on days 1, 8, 15, and 22	Phase II clinical trial, open label study, chemotherapy-naive CRPC, ECOG ≤2^−^, *n* = 26	PSA↓(*n* = 4) (no changes to PSA doubling time (, median OS → 13 months, no adverse impact on QoL, modest biochemical responses (*n* = 14) → minimal activity of Temsirolimus, SD→ majority of patients	Common ADEs: anemia	Median time: 32 months (range: 2–37)	[108]
Ridaforolimus (30 mg/day, oral) + bicalutamide (50 mg/day, oral)	Multicenter safety lead-in trial, open-label, progressive asymptomatic mCRPC, ECOG ≤1, *n* = 12 (11 patient discontinued)	↓30% PSA (*n* = 4), no pharmacokinetic differences in combination therapy, the impact of the AR blockade on the toxicity of mTOR inhibitors was not affected by PK interaction	Hypercholesterolemia, pneumonitis (grade 4), cheilitis, palmar-plantar, erythrodysesthesia, syndrome, edema, hyperglycemia, hypophosphatemia, and dehydration	Median time: 14.8 weeks (range 3.4–56.0 weeks)	[98]
Everolimus (10 mg, daily, oral)	Phase II clinical trial, single-agent, Simon’s two-stage design, chemotherapy-naive mCRPC with progressive disease (PCWG2 criteria), ECOG < 2	The median PFS → 2.8 mo, SD→*n* = 9, modest activity of everolimus, ↓the numbers of CD3, CD4, and CD8 T cells, T lymphocytes, ↑regulatory T cells p values ≤ 0.005 95% CI	Stomatitis, skin toxicities, pain, lymphopenia, pneumonitis, ↑ aspartate, aminotransferase, and creatinine	Median time: 16 weeks (range: 0.3–48)	[114]
Ridaforolimus (50 mg, i.v., once weekly)	Phase II trial, open-label, non-randomized, single-arm, Taxane-treated CRPC patients, *n* = 39 (all discontinued)	No objective responses, SD (*n* = 18), median time to progression: 28 days, PD (*n* = 5), ridaforolimus → well tolerated, PSA levels↑ (*n* = 23) 95% CI	Thrombocytopenia neutropenia, hypertriglyceridemia, hyperglycemia, hypokalemia, mucosal inflammation, hypercholesterolemia, pneumonitis, anemia, and peripheral edema	Median time: 109.5 days (range, 1–442 days)	[97]
Ipatasertib (200–400 mg, oral) or placebo + abiraterone (1000 mg, once daily, oral) + prednisone/prednisolone (5 mg, twice daily, oral)	Phase Ib/II trial, randomized 1:1:1, metastatic PC previously treated with docetaxel-based therapy and progressing after ≥1 hormonal therapy, ECOG 0–1, *n* = 253 - Ipat 400 mg group: 84 (66 patients discontinued) - Ipat 200 mg group: 87 (69 patients discontinued) -Placebo group: 82 (69 patients discontinued)	rPFS prolongation, well tolerance, superior antitumor activity in the Ipatasertib combination arms (Ipat 400 mg group) (especially PTEN-loss patients), PSA progression: placebo group > Ipat 200 mg group> Ipat 400 mg group, OS improvement: Ipat 200 mg group > Ipat 400 mg > placebo group	Hematuria, urinary retention, pyrexia, anemia, urinary tract in sepsis, septic shock, rash, and pain	24 months	[120]
Abiraterone acetate once daily (1000 mg, oral) + prednisone twice daily (5 mg, oral) + buparlisib once daily (100 mg or 60 mg, oral)	Phase Ib dose-finding study, open-label¸ patients with CRPC, failed Abiraterone Acetate treatment, ≤2 lines of prior chemotherapy (i.e., docetaxel, ECOG ≤ 2, Buparlisib 100 mg) cohort: *n* = 20 Buparlisib 60 mg cohort: *n* = 5	↓>30% in the PSA (*n* = 1), the MTD/RDE was not reached, no clinically meaningful antitumor activity, no loss of PTEN expression, stable disease (*n* = 3) (100 mg Buparlisib)	Hyperglycemia, asthenia, anemia, hypophosphataemia, and asthenia	Buparlisib (100 mg) cohort: 12.1 weeks Buparlisib (60 mg) cohort: 12 weeks	[121]
Abiraterone acetate once daily (1000 mg, oral) + prednisone twice daily (5 mg, oral) + dactolisib twice daily (200 mg, oral)	Phase Ib dose-finding study, open-label, patients with CRPC, failed Abiraterone Acetate treatment, ≤2 lines of prior chemotherapy (i.e., docetaxel, ECOG ≤ 2), *n* = 18	The MTD/RDE was not reached, marginal antitumor activity, challenging safety and tolerability profile	Asthenia, anemia, lymphopenia, thrombocytopenia, ↑ blood alkaline phosphatase and bone pain	The median time: 14.9 weeks (range, 1.9–74.0).	[121]

Abbreviations: ↓ = decrease; ↑ = increase; → = leads to; ADEs = adverse effects; AUC = area under the curve; BKM120 = buparlisib; CI = confidence interval; CR = complete response; CTCs = circulating tumor cells; DLTs = recommended dose for expansion; FDA = Food and Drug Administration; FDG = fluorodeoxyglucose; KPS = Karnofsky Performance Status; mCRPC = metastatic castration-resistant prostate cancer; MTD = maximum tolerated dose; NCAD = N-cadherin; OR = overall response; OS = overall survival; PCWG2 = Prostate Cancer Working Group 2; PD = progressive disease; PR = partial response; PSA = prostate specific antigen; PSF = progression-free survival; RECIST = response evaluation criteria in solid tumors; RDE = recommended dose for expansion; rPFS = radiographic progression-free survival; TCGA = the cancer genome atlas; SAE = serious adverse event; SUV = standardized uptake value; and SD = stable disease.

## 8. Preclinical Studies on Natural Agents Targeting mTOR-Signaling in Prostatic Neoplasms

To date, a number of plant-derived natural products have been shown to have favorable activities against prostatic neoplasms through the down-regulation of the mTOR-signaling pathway. With this understanding, future studies should target natural and/or synthetic components that are able to interfere with mTOR and/or its associated pathways. Clinical studies should be further conducted to clarify the potential therapeutic role of mTOR in prostatic neoplasms patients. The variety of actions of mTOR in the modulation of intestinal inflammatory events underscores a novel receptor targeting for the management of prostatic neoplasms (Table 4 and Table 5). In following sections, natural agents that target the mTOR-signaling in PC are introduced.

### 8.1. Apigenin

Apigenin (4,5,7-trihydroxyflavone) is a plant-derived polyphenol with significant anticancer effects [122]. Apigenin treatment decreased the viability of CD44 + PC stem cells via induction of the extrinsic apoptosis pathway by increasing caspase-8, caspase-3, PI3K, Akt, TNF-α, and Bcl-2 expression. Apigenin could not change the expression levels of genes involved in multidrug resistance such as ABCB1 and ABCC1. In PC3 cells, apigenin or a combination of this compound and docetaxel induced the intrinsic apoptosis pathway. In both PC stem cells and PC3 cells, apigenin elevated the expression of genes relevant to the progression of cell development, such as p21 and p27. Apigenin also inhibited the migration of PC stem cells [123].

**Table 4 pharmaceutics-13-01195-t004:** In vitro studies on natural PI3K/Akt/mTOR inhibitors in PC.

Treatment	Cell Line Name	Mechanism of Action	Results	Reference
Rottlerin (0.5–2 µM)	Prostate cancer stem cells	Rottlerin induces apoptosis by inhibiting the PI3K/Akt/mTOR pathway	↓phosphorylated Akt and mTOR cleavage of caspase-3 and caspase-9 ↓Bcl-2 and Bcl-xL protein levels ↑Bax level	[124]
MSeA (5, 10 µmol/L)	PC3 and PC3 M	mTOR inhibition in PC cells caused partial resistance to MSeA-induced growth reduction in hypoxia	In PC3 M: inhibition of HIF-1α in the absence of Rapamycin and after longer duration (6 h) in the presence of it in hypoxia MSeA + rapamycin: ↓REDD1 and pAkt &pRPS6 levels, modest resistance to growth inhibition in hypoxia In PC3 and PC3 M: growth inhibition in hypoxia	[125]
Shikonin (0.5–2 μM)	PC3 and DU145	Shikonin inhibited MMP-2/9 expression through attenuating the Akt/mTOR pathway	Inhibition of expression and activation of MMP-2 and MMP-9 → ↓metastasis ↑ expression of Bax, Bcl-2 ↓ expression of cyclin D1 ↓ phosphorylation levels of Akt and mTOR	[126]
Royleanone (12.5 µM)	LNCaP	↓ phosphorylation of PI3K, mTOR, and Akt→↓ mTOR/PI3K/Akt signaling pathway	↓ apoptosis ↓ G0/G1 cell cycle ↓ cell migration	[127]
Ursolic acid (80 Μm)	LNCaP and PC3	↓ expression of PI3K→↓phosphorylation of Akt→↓ mTOR-signaling proteins	↓ proliferation ↓ apoptosis ↓ cell growth	[128]
Afrocyclamin A (8 µM)	DU145	↓ phosphorylation of PI3k and Akt→↓mTOR activation	↑ cytotoxic ↑ apoptosis ↑ cell death ↓ invasion and migration	[71]
DHA (10 µM) and NAC (5 mM)	PC3 and DU145	DHA: ↓phosphorylated Akt and mTOR NAC: ↑phosphorylated Akt and mTOR	ROS-mediated Akt-mTOR signaling →autophagy and apoptosis induced by DHA in p53-mutant pc cells	[129]
DHA (10 µM)	PC3 and DU145	↑ number of cells with Sub-G1 DNA content, which represents hypodiploid nuclei (in PC3 cells)—↑ TUNEL positive cells and cleaved PARP ↑expression l of LC3-II ↑intracellular ROS	↓ viability and apoptosis in p53-mutant PC cells ↑ autophagic membranes ↑ autophagy and apoptosis by triggering intracellular ROS accumulation and DHA-induced intracellular ROS accumulation originated from mitochondria	[129]
Resveratrol (100 μM)	DU145 and PC3	↓ STIM1 expression →↓ interaction of STIM1 with TRPC1/Orai1 →↓ Ca entry → ↓Akt/mTOR pathway signaling → cell death	↓ cell proliferation and survival ↓STIM1 ↓ Akt1 phosphorylation	[130]
Baicalein (20, 40 μM)	DU145 and PC3	↑ Bax/Bcl-2 ratio → ↑ apoptosis PARP cleavage → cell apoptosis inhibition of the caveolin-1/Akt/mTOR pathway → ↓ cell proliferation	↓ proliferation ↑ apoptosis ↑ Bax level ↓ Bcl-2 ↓ survivin expression ↑ PARP cleavage ↓ cell migration ↓ Caveolin-1 expression ↓ phosphorylation of Akt and mTOR	[131]
Baicalein (2.5–20 μg/mL)	PC3 and DU145	↓ Cdk→ ↓ cell cycle arrest ↓ Mcl-1→ ↑ activation caspase-3, caspase-7 and poly(ADP-ribose) polymerase (PARP) cytochrome c release from mitochondria	↑ cell cycle arrest ↑ sub-G0cell population ↓ cell cycle regulators (cdc2, cdk2, cdk4, and cyclin D1) ↑p21 levels↑ ↓ protein levels and mRNA expression of mTOR and Raptor in PC3 ↑ autophagy Activation of AMPK/ULK1/mTORC1	[132]
Quercetin (10–50 μmol/L) + VEGF (10 ng/mL)	HUVECs	Quercetin: ↓ VEGFR2 downstream signaling molecules (Akt, mTOR, and p70S6K), ↓phospho-Akt (Ser473)	↓ HUVECs proliferation, chemotactic migration, and invasion ↓ angiogenesis through direct inhibition of VEGFR2 ↓ phospho-mTOR levels and phospho-Akt and phospho-S6K	[133]
Quercetin (10–50 μmol/L)	PC3	Quercetin: activation of VEGFR2 downstream signaling molecules (Akt, mTOR, and p70S6K)↓, phospho-Akt (Ser473)↓	↓ VEGF secretion ↑ apoptosis and Akt/mTOR/P70S6K pathway ↓ cell viability	[133]
Resveratrol (2.5–10 μM) with or without 2 and 8 Gy ionizing radiation	PC3 and 22RV1	↑ ATM-AMPK-p53-p21cip1/p27kip1 signaling pathway ↓ Akt signaling pathways ↑ substrate histone H2Ax ↑ expression of p53 & CDKIs	↓ cell survival ↓ dose of radiotherapy ↓ IR-mediated cell cycle arrest (at G2/M interphase) ↑ cell accumulation at G1/S and sub-G1 phases ↑ caspase-3 cleavage ↑ IR-induced nuclear aberrations ↓ IR-induced levels of phosphorylated Akt on S473 and Akt T308 phosphorylation	[134]
DHA (40 μM) and AA (40 μM)	LNCaP	DHA: ↓ Akt/mTOR-signaling pathway, ↓ Akt1 activity, ↓ phospho-S6 levels AA: ↑ Akt activity levels, ↑ phospho S6 levels	DHA: ↓ phospho-Akt and phospho-tuberin levels ↓ mTOR pathway in DHA-treated cells ↓ androgen receptor expression (long-term treatment) AA: ↑ phospho-Akt and phospho-tuberin levels Minimal effect on suppressing androgen deprivation-induced expression of AR	[135]
DHA (20 μM), EPA (20 μM), and AA (20 μM)	LNCaP	Omega-3 fatty acids (DHA and EPA): ↓ Akt/mTOR-signaling pathway ↓ Akt1 activity ↓ phospho-S6 levels AA: ↑ Akt activity levels ↑ phospho S6 levels	Omega-3 fatty acids (DHA and EPA): ↓ progression and cell proliferation (conditions of hormone depletion), expression of the androgen receptor AA: ↑ androgen-independent proliferation ↑ cell growth (conditions of hormone depletion)	[135]
Aloe-emodin (2.5–15 μM)	PC3	↓ activation of the downstream substrates of mTORC2, Akt, and PKCa	↓ cell proliferation ↓ anchorage-independent growth of PC3 cells ↓ mTORC2-mediated downstream signaling (Akt phosphorylation at Ser473 and GSK3β at Ser9) ↓ phosphorylation of Akt1 at Ser473 mTOR kinase activity was not be inhibited	[136]
DHA (50 µM) and NAC (5 mM)	PC3 and DU145	DHA: ↓ phosphorylated Akt and mTOR NAC: ↑ phosphorylated Akt and mTOR	ROS-mediated Akt-mTOR signaling → autophagy and apoptosis induced by DHA in p53-mutant pc cells	[129]
DHA (50 µM)	PC3 and DU145	↑ number of cells with Sub-G1 DNA content, which represents hypodiploid nuclei (in PC3 cells)—↑ TUNEL positive cells and cleaved PARP ↑ expression l of LC3-II ↑ intracellular ROS	↓ viability and apoptosis in p53-mutant PC cells ↑ autophagic membranes ↑ autophagy and apoptosis by triggering intracellular ROS accumulation and DHA-induced intracellular ROS accumulation	[129]
Resveratrol (100 μM)	DU145 and PC3	↓ STIM1 expression →↓interaction of STIM1 with TRPC1/Orai1 →↓Ca entry → ↓Akt/mTOR pathway signaling → cell death	↓ cell proliferation and survival ↓ STIM1 ↓ Akt1 phosphorylation	[130]
Rottalin (2 μM)	PC3 and DU145	↓ phosphorylation of P70S6K and S6→ ↓mTORC 1 signaling	↓ Cytosolic β-catenin level ↓ expression of axin2 and LRP	[137]

Abbreviations: ↓ = decrease; ↑ = increase; → = leads to; AA = arachidonic acid (omega-6 fatty acid); Bcl-2 = B-cell lymphoma 2; DHA = docosahexaenoic acid; EPA = eicosapen taenoic acid (omega-3 fatty acid); ERR = ethanol extract of *Remotiflori radix*; HUVEC = human umbilical vein endothelial cell; HIF-1α = hypoxia-inducible factor; LRP6 = low-density lipoprotein receptor protein 6; Mcl-1 = myeloid cell leukemia 1; MMP-2 = matrix metalloproteinase-2; MSeA = methylseleninic acid; NAC = N-acetyl-cysteine; PARP = poly(ADP-ribose) polymerase; PDK1 = 3-phosphoinositide-dependent protein kinase-1; PKC = protein kinase C; STIM1 = stromal interaction molecule 1; and VEGF = vascular endothelial growth factor.

**Table 5 pharmaceutics-13-01195-t005:** In vivo studies on natural PI3K/Akt/mTOR inhibitors in prostate cancer.

Treatment	Animals	Model	Number of Animals	Treatment Duration	Results	Reference
Quercetin (20 mg/kg/day, i.p.)	Rat aortic ring	Angiogenesis study model	*n* = 6	6 days	↓micro-vessel growth ↓angiogenesis	[133]
Quercetin (20 mg/kg/day, i.p.)	Male BALB/c nude mice	PC3 tumor xenograft	*n* = 3	15 days	↓tumor angiogenesis ↓tumor growth (tumor volume and tumor weight)	[133]
Afrocyclamin A (5 mg/kg, i.p.) and Doxorubicin (2 mg/kg, i.p.)	Male nude mice	DU145 tumor xenograft	*n* = 6	4 weeks	↓tumor volume and weight ↓intensity of Ki67 ↑apoptosis ↑cell death	[71]
Aloe-emodin (10, 50 mg/kg, i.p.)	BALB/c nude mice	PC3 tumor xenograft	*n* = 12	30 days	↓phosphorylation of Akt1 at Ser473 ↓tumor growth and weight ↓mTORC2	[136]
*Remotiflori radix* (50 mg/kg, oral)	BALB/c nude mice	PC3 and DU-145 tumor xenograft	*n* = 55	4 weeks	↓tumor size	[86]
Ursolic acid (20 or 40 mg/kg, i.p.)	Female athymic nude	LNCAPs tumor xenograft	*n* = 3	3 weeks	↓proliferation ↓apoptosis ↓tumor size ↓mTOR activity through PI3K/Akt/mTOR ↓phosphorylation of Akt	[128]
Piperlongumine (20 mg/kg, daily, i.p.); Chloroquine (40 mg/kg, daily, i.p.); and Piperlongumine (20 mg/kg, daily, i.p.) + chloroquine (40 mg/kg, daily, i.p.)	Male C.B17/Icr-scid mice	PC-3 tumor xenograft	*n* = 5	Not mentioned	↓tumor growth and tumor mass	[138]

Abbreviations: ↓ = decrease; ↑ = increase; → = leads to; CAF = cancer-associated fibroblasts; CQ = chloroquine; KO = knockout; PC = prostate cancer; PL = piperlongumine; ROS = reactive oxygen species; TGF-β = transforming growth factor-β; and SMA = smooth muscle actin.

### 8.2. Arctigenin

Arctigenin, a natural phenyl propanoid dibenzyl butyrolactone lignin isolated from seeds of Arctium lappa, was shown to hamper PC [139]. Arctigenin enhanced cell death in PC3 AcT cells, while increasing G1 and S arrest. Similar outcomes were obtained when arctigenin was used in combination with docetaxel [140]. Arctigenin elevated the cellular ROS level and increased the cytotoxicity and mitochondrial membrane potential loss in PC3 cells. Moreover, arctigenin-induced ROS accumulation was implicated in the ATP depletion and suppression of the PI3K/Akt/mTOR pathway [140].

### 8.3. Docosahexaenoic Acid (DHA)

DHA is an omega-3 polyunsaturated fatty acid (ω3-PUFAs) that is able to decrease the risk of PC, unlike saturated fatty acids [141]. Treatment of PC3 and DU145 cells with DHA enhanced apoptosis and autophagy in these cells, which was attributed to the activation of the mitochondrial ROS-mediated Akt-mTOR-signaling pathway [129].

### 8.4. Piperlongumine

Piperlongumine is an alkaloid isolated from the Long pepper (*Piper longum*) [142]. In androgen-independent PC3 cells, piperlongumine enhanced apoptosis through deactivation of the Akt/mTORC1 signaling pathway. Inhibition of Akt by piperlongumine resulted in the suppression of the mTORC1 complex [138].

### 8.5. Resveratrol

Resveratrol (3,4′-trihydroxystilbene) is a natural stilbene present in the skins of grapes, peanuts, and red wine [143] with several anticancer, anti-inflammatory, and antioxidant properties. In LNCaP, C42B, RWPE-1, and DU145 cells treated with resveratrol, the phosphorylation of S6K was prevented but the phosphorylation of Akt (T308 or S473) did not change (the Akt/mTOR-signaling components). Resveratrol also induced autophagy in PC cells through suppression of S6K phosphorylation mediated by SIRT1 (kind of histone) [144,145]. Additionally, resveratrol prevented PIN lesion growth in vivo through the inhibition of the Akt/mTOR-signaling pathway. Resveratrol also caused a reduction in staining for pS6K and an enhancement of staining for SIRT1 without any change in the expressions of phosphor-Akt or phospho-mTOR in PTEN knocked out mice [137].

### 8.6. Tangeretin

Tangeretin (4′,5,6,7,8-pentamethoxyflavone) is a flavonoid with known anticancer activity in breast, colorectal, lung, and gastric carcinoma [146]. The compound was shown to reduce prostate cancer cells’ (PC3 and LNCaP) viability through a time and concentration-dependent manner by the suppression of Bcl-2 and induction of apoptosis. Tangeretin restrained the ability of PC3 cell colony formation, restrained the mobility of cancer cells, and suppressed the Akt signaling [147].

### 8.7. 3-Cinnamoyl-KβBA

3-Cinnamoyl-11-keto-β-boswellic acid (cinnamoyl-KβBA) is a semisynthetic triterpenoid compound formed by the addition of cinnamic acid to KβBA. It was shown that C-KβBA reduced the proliferation of PC cells via the induction of apoptosis in vitro and in PC3 xenografts in vivo. C-KβBA suppressed the phosphorylation of p70 ribosomal S6 kinase, a downstream target of mTOR complex 1. However, C-KβBA strongly binds to the FKBP12-rapamycin-binding domain of mTOR and thus this compound might act as a proapoptotic mTOR inhibitor [46,148].

### 8.8. Miscellaneous Compounds

*Ocimum sanctum* Linn from Lamiaceae is a popular medicinal plant with multiple biological and pharmacological properties including anticancer effects [149]. Among various flavonoids of *O. sanctum*, orientin, luteolin and vicenin-2 (VCN-2), VCN-2 was found to be the most effective in prevention of PC [150,151]. In PC cells, VCN-2 effectively prevented angiogenesis, although it could not inhibit PC cells’ migration. VCN-2 enhanced the poly (ADP-ribose) polymerase (PARP) cleavage and expression of Bax, and reduced the Bcl-2 expression, decreased the amount of pAkt, and suppressed the pP70S6K activity. A combination of VCN-2 and docetaxel synergistically reduced the viability of PC cells. In xenograft mouse models, VCN-2 or docetaxel administration resulted in a reduction in tumor weight and positive synergism was observed when VCN-2 and docetaxel were combined [151].

## 9. Conclusions

PC is a serious public health issue, representing the third highest cause of mortality worldwide, especially in developed countries. Various factors are involved in the pathophysiology of PC including human papillomavirus, cytokines, or a nutritional regimen. Multiple studies exhibited that the PI3K-Akt-mTOR pathway could potentially play an important role in PC as a therapeutic target and/or a predictive biomarker for the onset, progression, and behavior of the disease. Additionally, the TGF-β/PI3K/Akt-mTOR-NF-κB transduction pathway has been confirmed to be activated in PC. High expression of the PI3K/Akt/mTOR pathway in PC represents the critical role of this pathway in PC progression. Clinical trials exhibited that the inhibitors of the PI3K/Akt/mTOR pathway are novel targets for treatment of PC. Thus, this review presents strong evidence to introduce the P13K/Akt/mTOR inhibitors as plausible therapeutic targets for PC. Future clinical trials should focus on targeting the signaling of PI3K, mTORC1/2, and androgen to increase the survival rate and improve the life quality in PC patients. Current preclinical and clinical data indicate that inhibitors of the PI3K/Akt/mTOR pathway in combination with other anticancer therapies might have greater utility to suppress the progression of PC and its resistance to chemotherapy.

## Figures and Tables

**Figure 1 pharmaceutics-13-01195-f001:**
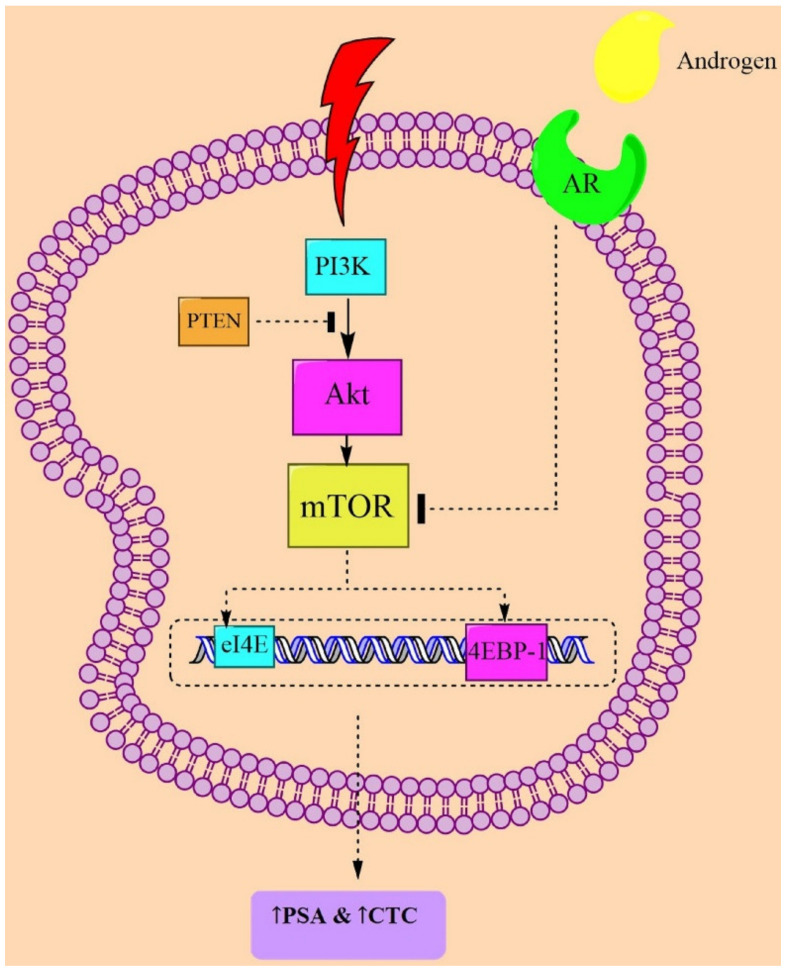
Possible mechanisms from clinical studies targeting mTOR in PC pathogenesis. Abbreviations: AR = androgen receptor; CTC = circulating tumor cell; mTOR = mammalian target of rapamycin; PTEN = phosphatase and tensin homolog; 4EBP1 = 4E-binding protein 1; PSA = prostate-specific antigen; and PI3K = phosphoinositide 3-kinase.

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
