# Peer review of "Inhibitors of the PI3K/Akt/mTOR Pathway in Prostate Cancer Chemoprevention and Intervention"

_pharmaceutics, 2021, doi:10.3390/pharmaceutics13081195_

Round 1
Reviewer 1 Report
Prostate cancer is one of the most common forms of cancer in men, and the treatment involves surgical, radiation, or hormonal therapy. In this review, the authors point it out the usefulness of interfering with PI3K/Akt/mTOR pathway and the clinical implications of the use of mTOR inhibitors as therapeutic agents.
The topic is interesting and relevant, but I have some comments to the authors:
Page 2 line 61. There is no connection between the sentence about corticosteroids and the rest of the paragraph. Modify the way you approach the use of corticosteroids in prostate cancer treatment.
Page 3 line 112: the word “PRAS40” is written twice.
Page 4 line 168 and Page 5 lines 186 and 187: the word “PTEN” is first used on page 4, but what the abbreviation means is written on page 5.
Page 5: “5. Prostatic Neoplasms and PI3/Akt/mTOR Signaling”. I would rearrange the text into one section (without the subheadings), highlighting the most important findings regarding the dysregulation of PI3/Akt/mTOR signaling in prostate cancer.
Page 6 lines 257-264: It is not clear why C-KβBA and FGF23 are in the same subheading.
Table 5 and Fig 3: Fig 3 covers the last part of table 5. It may be that this is a problem that arose when the pdf was generated.
Author Response
The authors of this manuscript express their sincere thanks to the reviewer for the critical assessment of this work. The authors have acted upon the recommendations of the editors and the reviewers which have resulted in a significant enhancement in the quality of this manuscript. All modifications incorporated in the manuscript are highlighted in red color font. A “point-by-point” response to each and every comment is outlined below:
General comments:
Prostate cancer is one of the most common forms of cancer in men, and the treatment involves surgical, radiation, or hormonal therapy. In this review, the authors point it out the usefulness of interfering with PI3K/Akt/mTOR pathway and the clinical implications of the use of mTOR inhibitors as therapeutic agents.
The topic is interesting and relevant, but I have some comments to the authors:
Response:
We thank the reviewer for their expertise, time, and effort for reviewing our manuscript. We are deeply encouraged by the generous comments about the quality of our work. As described below, we have revised our manuscript based on the reviewer’s specific comments.
Specific comments:
Comment 1:
Page 2 line 61. There is no connection between the sentence about corticosteroids and the rest of the paragraph. Modify the way you approach the use of corticosteroids in prostate cancer treatment.
Response:
Thanks for your valuable comment. We have revised the text to promote clarity (page 2, lines 61-66).
Comment 2:
Page 3 line 112: the word “PRAS40” is written twice.
Response:
We appreciate the reviewer’s watchful eyes. The duplicate word has been deleted (page 3, line 113).
Comment 3:
Page 4 line 168 and Page 5 lines 186 and 187: the word “PTEN” is first used on page 4, but what the abbreviation means is written on page 5.
Response:
Thanks for your precious comment. We have included the full form in our revised manuscript (page 2, line 92). We have corrected subsequent text accordingly (page 4, line 167, and page 5, line 187).
Comment 4:
Page 5: “5. Prostatic Neoplasms and PI3/Akt/mTOR Signaling”. I would rearrange the text into one section (without the subheadings), highlighting the most important findings regarding the dysregulation of PI3/Akt/mTOR signaling in prostate cancer.
Response:
Thanks for your worthy comment. We omitted the subheadings and the text has been merged into one section. Only important parts have been included (page 5, line 199 to page 6, line 246).
Comment 5:
Page 6 lines 257-264: It is not clear why C-KβBA and FGF23 are in the same subheading.
Response:
We appreciate this comment. The text has been separated (page 6, lines 233-242 and page 22, lines 490-497).
Comment 6:
Table 5 and Fig 3: Fig 3 covers the last part of table 5. It may be that this is a problem that arose when the pdf was generated.
Response:
According to the comments of Reviewer 2, the quality of Figure 3 has been improved, and now it is represented as Figure 2.
Additionally,
- The reference list has been modified and renumbered accordingly. Special attention is given to conform to the order of references and bibliographic style of the journal.
- We have deleted a figure (previously Figure 2) based on the reviewer’s comment. Previous Figure 3 has become Figure 2.
- The entire manuscript has been thoroughly checked and edited to ensure uniform style, organization, and quality.
Finally,
On behalf of my co-authors, I once again express my sincere thanks to the erudite reviewer for the valuable suggestions and constructive input to improve the quality of our manuscript.
Reviewer 2 Report
The review pharmaceutics-1297301, Inhibitors of the PI3K/Akt/mTOR Pathway in Prostate Cancer Chemoprevention and Intervention, has a fairly good design, but it could better organized. The English language and style need some corrections. The subject of the paper is well suited for the journal and could be of its readers’ interest if more focus is put on the pharmaceutical view of the subject.
The number of references could have been higher and more recent, considering the number of pages and the volume of data in this field. I advise the authors to present and reference other review works that present similar information and to highlight their own contribution compared to existing ones. Here is a list of very similar review works that should be referenced and discussed.
- Mechanisms of Resistance to PI3K Inhibitors in Cancer: Adaptive Responses, Drug Tolerance and Cellular Plasticity, Cancers, 2021 Mar 26;13(7):1538
- The PI3K-AKT-mTOR Pathway and Prostate Cancer: At the Crossroads of AR, MAPK, and WNT Signaling.
- The Akt pathway in oncology therapy and beyond (Review). Int. J. Oncol. 2018, 53, 2319–2331.
- Yang, J., Nie, J., Ma, X. et al. Targeting PI3K in cancer: mechanisms and advances in clinical trials. Mol Cancer 18, 26 (2019)
- Akt inhibitors in cancer treatment: The long journey from drug discovery to clinical use (Review) Int J Oncol. 2016 Mar;48(3):869-85
I hope these reviews could be a good model for the authors, considering that they present a lot of substances that are somehow connected with the PI3K/Akt/mTOR pathway, but are not really targeted drug against any of these protein kinases.
The sections 6 and 8 are mostly wrong in my view. I advise the authors to describe only actual inhibitors of the PI3K or Akt or mTOR, and not drugs that indirectly could influence these. For example, buparlisib, dactolisib, or ipatasertib. Remove drugs like Abiraterone, Cyclosporine A, Cabazitaxel, Docetaxel. The section on morpholino thienopyrimidine adds little real information. The referenced articles are over 10 years old. Are there relevant new data on a specific compound that was used in a clinical trial on prostate cancers? The authors should remove any similar compounds that are not relevant.
The table 1 should also be changed accordingly. The table is poorly organized. There some drugs that are presented in many rows without a clear motive to do so. Again, the authors should focus only on drugs that directly target this pathways. Drugs that are proven to inhibit PI3K, or inhibitors of Akt, or direct inhibitors of mTOR. See for example, CmpdA. It is a IKKβ inhibitor. There is no clear reason to present it as PI3K/Akt/mTOR. It can be really confusing for the readers and it could induce wrong knowledges.
Remove the figure 2. The presented structures are not relevant and the graphical quality is poor. The same advice is for figure 3.
The authors should detail the FDA and EMA approval status for the drugs targeting PI3K or Akt or mTOR the authors should add the approved indications.
Author Response
The authors of this manuscript express their sincere thanks to the reviewer for the critical assessment of this work. The authors have acted upon the recommendations of the editors and the reviewers which have resulted in a significant enhancement in the quality of this manuscript. All modifications incorporated in the manuscript are highlighted in red color font. A “point-by-point” response to each and every comment is outlined below:
General comments:
The review pharmaceutics-1297301, Inhibitors of the PI3K/Akt/mTOR Pathway in Prostate Cancer Chemoprevention and Intervention, has a fairly good design, but it could better organized. The English language and style need some corrections. The subject of the paper is well suited for the journal and could be of its readers’ interest if more focus is put on the pharmaceutical view of the subject.
Response:
We would like to thank the erudite reviewer for his/her appreciation and critical assessment of the manuscript with constructive suggestions. We have tried our best to improve the quality of our manuscript based on the reviewer’s specific comments. Additionally, our manuscript has been edited thoroughly.
Specific comments:
Comment 1A:
The number of references could have been higher and more recent, considering the number of pages and the volume of data in this field. I advise the authors to present and reference other review works that present similar information and to highlight their own contribution compared to existing ones. Here is a list of very similar review works that should be referenced and discussed.
- Mechanisms of Resistance to PI3K Inhibitors in Cancer: Adaptive Responses, Drug Tolerance and Cellular Plasticity, Cancers, 2021 Mar 26;13(7):1538
- The PI3K-AKT-mTOR Pathway and Prostate Cancer: At the Crossroads of AR, MAPK, and WNT Signaling.
- The Akt pathway in oncology therapy and beyond (Review). Int. J. Oncol. 2018, 53, 2319–2331.
- Yang, J., Nie, J., Ma, X. et al. Targeting PI3K in cancer: mechanisms and advances in clinical trials. Mol Cancer 18, 26 (2019)
- Akt inhibitors in cancer treatment: The long journey from drug discovery to clinical use (Review) Int J Oncol. 2016 Mar;48(3):869-85
Response:
We admire the reviewer for this excellent suggestion. We have discussed the suggested publications as follows:
- Mechanisms of Resistance to PI3K Inhibitors in Cancer: Adaptive Responses, Drug Tolerance and Cellular Plasticity, Cancers, 2021 Mar 26;13(7):1538 – page 5, line 188 (Ref. No. 43).
- The PI3K-AKT-mTOR Pathway and Prostate Cancer: At the Crossroads of AR, MAPK, and WNT Signaling - page 2, line 75 (Ref. No. 8).
- The Akt pathway in oncology therapy and beyond (Review). Int. J. Oncol. 2018, 53, 2319–2331 - page 5, line 192 (Ref. No. 44).
- Yang, J., Nie, J., Ma, X. et al. Targeting PI3K in cancer: mechanisms and advances in clinical trials. Mol Cancer 18, 26 (2019) - page 2, line 90 (Ref. No. 11).
- Akt inhibitors in cancer treatment: The long journey from drug discovery to clinical use (Review) Int J Oncol. 2016 Mar;48(3):869-85 - page 3, line 100 (Ref. No. 13).
Comment 1B:
I hope these reviews could be a good model for the authors, considering that they present a lot of substances that are somehow connected with the PI3K/Akt/mTOR pathway, but are not really targeted drug against any of these protein kinases.
Response:
Thanks for your precious comment. As mentioned before, we have added the recommended references. Based on the excellent suggestion, discussion on drugs that targeted mTOR indirectly has been deleted (page 2, line 69 to page 3, line 105; and page 5, line 199 to page 6, line 246).
Comment 2:
The sections 6 and 8 are mostly wrong in my view. I advise the authors to describe only actual inhibitors of the PI3K or Akt or mTOR, and not drugs that indirectly could influence these. For example, buparlisib, dactolisib, or ipatasertib. Remove drugs like Abiraterone, Cyclosporine A, Cabazitaxel, Docetaxel. The section on morpholino thienopyrimidine adds little real information. The referenced articles are over 10 years old. Are there relevant new data on a specific compound that was used in a clinical trial on prostate cancers? The authors should remove any similar compounds that are not relevant.
Response:
Thanks for your valuable comment. The text has been revised according to your suggestion. Irrelevant data has been omitted (page 6, line 249 to page 14, line 358 and page 17, line 406 to page 22. line 498). New references (Ref. No. 8, 10, 11, 12 13, 43, and 44) have been cited and discussed in our revised manuscript (page 2, line 75; page 2, line 92, page 3, line 102, page 5, line 188; page 5, line 192; page 2, line 90; and page 3, line 100).
Comment 3:
The table 1 should also be changed accordingly. The table is poorly organized. There some drugs that are presented in many rows without a clear motive to do so. Again, the authors should focus only on this pathway. Drugs that are proven to inhibit PI3K, or inhibitors of Akt, or direct inhibitors of mTOR. See for example, CmpdA. It is a IKKβ inhibitor. There is no clear reason to present it as PI3K/Akt/mTOR. It can be really confusing for the readers and it could induce wrong knowledge.
Response:
We agree with this excellent comment. Accordingly, Table 1 (page 6) has been revised. Irrelevant drugs have been omitted. Only drugs that directly target PI3K/Akt/mTOR are presented ).
Comment 4:
Remove the figure 2. The presented structures are not relevant and the graphical quality is poor. The same advice is for figure 3.
Response:
Thanks for your comment. Figure 2 has been removed and a better quality Figure 3 (now Figure 2, page 22) has been provided.
Comment 5:
The authors should detail the FDA and EMA approval status for the drugs targeting PI3K or Akt or mTOR the authors should add the approved indications.
Response:
This is an excellent suggestion. The FDA & EMA approval status for the drugs has been added to our revised manuscript (page 4, line 136; page 6, line 273; and page 13, line 333).
Additionally,
- The reference list has been modified and renumbered accordingly. Special attention is given to conform to the order of references and bibliographic style of the journal.
- We have deleted a figure (previously Figure 2) based on the reviewer’s comment. Previous Figure 3 has become Figure 2.
- The entire manuscript has been thoroughly checked and edited to ensure uniform style, organization, and quality.
Finally,
On behalf of my co-authors, I once again express my sincere thanks to the erudite reviewer for the valuable suggestions and constructive input to improve the quality of our manuscript.
Round 2
Reviewer 2 Report
The authors improved their manuscript by applying most of the suggestions of the previous review. There are still some problems, especially in the editing of the manuscript.
Again, I advise the authors to remove the section on morpholino thienopyrimidine. It adds little real information and are not relevant clinical. These compounds are not presented in any tables demonstrating their little importance. Instead, the authors should add a new section (let’s say 6.3) on Ridaforolimus and other important substances that are presented in the tables.
The authors should use everolimus, and not RAD001. Everolimus is the INN of this compound. Please correct also in all the tables and figures (in same places is already presented as everolimus).
Remove JQ1 from table 1. It is not relevant for this pathway.
Avoid using abbreviation in the tables. See for example ESK242, BEZ235 or INK128. The tables should be easy to read without needing the whole paper. If abbreviations are still used, they should be explained in the footnotes of the table. The table should have a uniform editing style. Choose between “10 Nm Rapamycin” or “Rapamycin (10 Nm)”, but do not use both.
In figure 2, the authors presented tetrahydroquinazoline, but it should be derivatives of 7-aza-tetrahydroquinazoline. The structure should be corrected to show that the authors are discussing a series of compounds. The general structure should be drawn here. Also, remove docetaxel, cabazitaxel, abiraterone, and morpholino thienopyrimidines
Author Response
The authors of this manuscript express their sincere thanks to the reviewer for the critical assessment of this work. The authors have acted upon the recommendations of the reviewers which have resulted in a significant enhancement in the quality of this manuscript. All modifications incorporated in the manuscript are highlighted in red color font. A “point-by-point” response to each and every comment is outlined below.
Comment 1:
The authors improved their manuscript by applying most of the suggestions of the previous review. There are still some problems, especially in the editing of the manuscript.
Response:
We appreciate the reviewer’s time and effort in checking our revised manuscript. We have addressed the concerns and once again revised our manuscript as described below.
Comment 2:
Again, I advise the authors to remove the section on morpholino thienopyrimidine. It adds little real information and are not relevant clinical. These compounds are not presented in any tables demonstrating their little importance. Instead, the authors should add a new section (let’s say 6.3) on Ridaforolimus and other important substances that are presented in the tables.
Response:
We agree with the reviewer. The sections on morpholino thienopyrimidine have been removed. A new section on ridaforolimus has been added (Section 6.3, page 13, lines 324-335). It has been presented as ref. 69 in Table 1 (page 7) and as refs. 99 and 100 in Table 3 (page 17).
Comment 3:
The authors should use everolimus, and not RAD001. Everolimus is the INN of this compound. Please correct also in all the tables and figures (in some places is already presented as everolimus).
Response:
Thanks for your valuable comment. We have used everolimus in place of RAD001 throughout the manuscript (Section 6.1., page 6, line 263-27; Table 1).
Comment 4:
Remove JQ1 from table 1. It is not relevant for this pathway.
Response:
We are in absolute agreement with the reviewer and deleted it from Table 1.
Comment 5:
Avoid using abbreviation in the tables. See for example ESK242, BEZ235 or INK128. The tables should be easy to read without needing the whole paper. If abbreviations are still used, they should be explained in the footnotes of the table. The table should have a uniform editing style. Choose between “10 Nm Rapamycin” or “Rapamycin (10 Nm)”, but do not use both.
Response:
We appreciate this comment. We tried to avoid abbreviations where it was possible. Relevant footnotes have been added to the tables to provide the full forms of all abbreviations. The tables have been edited to have a uniform style and appearance.
Comment 6:
In figure 2, the authors presented tetrahydroquinazoline, but it should be derivatives of 7-aza-tetrahydroquinazoline. The structure should be corrected to show that the authors are discussing a series of compounds. The general structure should be drawn here. Also, remove docetaxel, cabazitaxel, abiraterone, and morpholino thienopyrimidines.
Response:
We have deleted the figure as per the reviewer’s previous suggestion.
On behalf of my co-authors, I once again express my sincere thanks to the erudite reviewer for the valuable suggestions and constructive input to improve the quality of our manuscript.
Round 3
Reviewer 2 Report
The authors improved their manuscript by applying most of the suggestions of the previous review.
The table should have a uniform editing style. It should be “Rapamycin (10 Nm)”. Please add as footnotes the abbreviations for drugs like ESK242, BEZ235 or INK128.
Author Response
Comment:
The authors improved their manuscript by applying most of the suggestions of the previous review.
The table should have a uniform editing style. It should be “Rapamycin (10 Nm)”. Please add as footnotes the abbreviations for drugs like ESK242, BEZ235 or INK128.
Response:
We are grateful to the reviewer for his/her attention to detail. We have extensively edited the tables to ensure a uniform style. The concentration for rapamycin has been corrected. We have included generic names for agents in addition to code names as appropriate. In a few cases, the IUPAC name has been used since generic names are not available. All acronyms have been included in the footnotes.
On behalf of my co-authors, I once again express my sincere thanks to the erudite reviewer for the valuable suggestions and constructive input to improve the quality of our manuscript.